

# Enriching scientific publications with interactive 3D PDF: an integrated toolbox for creating ready-to-publish figures

Axel Newe

Medical Applications Team, Method Park Engineering GmbH, Erlangen, Germany
Chair of Medical Informatics, Friedrich-Alexander Universität Erlangen-Nürnberg, Erlangen, Germany

## ABSTRACT

Three-dimensional (3D) data of many kinds is produced at an increasing rate throughout all scientific disciplines. The Portable Document Format (PDF) is the de-facto standard for the exchange of electronic documents and allows for embedding three-dimensional models. Therefore, it is a well-suited medium for the visualization and the publication of this kind of data. The generation of the appropriate files has been cumbersome so far. This article presents the first release of a software toolbox which integrates the complete workflow for generating 3D model files and ready-to-publish 3D PDF documents for scholarly publications in a consolidated working environment. It can be used out-of-the-box as a simple working tool or as a basis for specifically tailored solutions. A comprehensive documentation, an example project and a project wizard facilitate the customization. It is available royalty-free and for Windows, MacOS and Linux.

# INTRODUCTION

Throughout many scientific disciplines, the availability—and thus the importance—of three-dimensional (3D) data has grown in the recent years. Consequently, this data is often the basis for scientific publications, and in order to avoid a loss of information, the visualization of this data should be 3D whenever possible (*Tory & Möller*, *2004*). In contrary to that, almost all contemporary visualization means (paper printouts, computer screens, etc.) only provide a two-dimensional (2D) interface.

The most common workaround for this limitation is to project the 3D data onto the available 2D plane (*Newe*, *2015*), which results in the so-called "2.5D visualization" (*Tory & Möller*, *2004*). This projection yields two main problems: limited depth perception and objects that occlude each other. A simple but effective solution of these problems is interaction: by changing the projection angle of a 2.5D visualization (i.e., by changing the point of view), depth perception is improved (*Tory & Möller*, *2004*), and at the same time objects that had previously been occluded (e.g., the backside) can be brought to sight.

A means of application of this simple solution has been available for many years: the Portable Document Format (PDF) from Adobe (*Adobe*, *2014*). This file format is the

Corresponding author
Axel Newe, axel.newe@fau.de

de-facto standard for the exchange of electronic documents and almost every scientific article that is published nowadays is available as PDF—as well as even articles of the middle of the last century (*Hugh-Jones*, *1955*). PDF allows for embedding 3D models and the Adobe Reader (http://get.adobe.com/reader/otherversions/) can be used to display these models interactively.

Nevertheless, this technology seems not to have found broad acceptance among the scientific community until now, although journals encourage authors to use this technology (*Maunsell*, *2010*; *Elsevier*, *2015*). One reason might be that the creation of the appropriate model files and of the final PDF documents is still cumbersome. Not everything that is technically possible is accepted by those who are expected to embrace the innovation if the application of this innovation is hampered by inconveniences (*Hurd*, *2000*). Generally suitable protocols and procedures have been proposed by a number of authors before, but they all required a toolchain of at least three (*Kumar et al.*, *2010*; *Danz & Katsaros*, *2011*) or even four (*Phelps, Naeger & Marcovici*, *2012*; *Lautenschlager*, *2014*) different software applications and up to 22 single steps until the final PDF was created. Furthermore, some of the proposed workflows were limited to a certain operating system (OS) (*Phelps, Naeger & Marcovici*, *2012*), required programming skills (*Barnes et al.*, *2013*) or relied on commercial software (*Ruthensteiner & Heß*, *2008*). Especially the latter might be an important limiting factor which hampers the proliferation of the 3D PDF format in scientific publishing (*Lautenschlager*, *2014*; *Newe*, *2015*).

This article presents a comprehensive and highly integrated software tool for the creation of both the 3D model files (which can be embedded into PDF documents) and the final, ready-to-publish PDF documents with embedded interactive 3D figures. The presented solution is based on MeVisLab, available for all major operating systems (Windows, MacOS and Linux) and requires no commercial license. The source code is available but does not necessarily need to be compiled since binary add-on installers for all platforms are available. A detailed online documentation, an example project and an integrated wizard facilitate re-use and customization.

## BACKGROUND AND RELATED WORK

### The Portable Document Format

The Portable Document Format is a document description standard for the definition of electronic documents independent of the software, the hardware or the operating system that is used for creating or consuming (displaying, printing…) it (*Adobe*, *2008a*). A PDF file can comprise all necessary information and all resources to completely describe the layout and the content of an electronic document, including texts, fonts, images and multimedia elements like audio, movies or 3D models. Therefore, it fulfils all requirements for an interactive publication document as proposed by *Thoma et al.*, *(2010)*.

Although it is an ISO standard (ISO 32000-1:2008 (*ISO*, *2008*)), the specification is available to the full extent from the original developer Adobe (*Adobe*, *2015*) and can be used royalty-free.

**Table 1** Number of publications related to 3D PDFs in biomedical sciences since 2008 (not comprehensive).

| Year | Number of publications with embedded/supplemental 3D PDF | Number of publications dealing with/mentioning 3D PDF |
|---|---|---|
| 2005 | – | 1 |
| 2008 | 1 | – |
| 2009 | 5 | 4 |
| 2010 | 2 | 7 |
| 2011 | 7 | 6 |
| 2012 | 6 | 5 |
| 2013 | 7 | 2 |
| 2014 | 21 | 7 |
| 2015 | 31 | 2 |

## Embedding 3D models into PDF

The fifth edition of the PDF specification (PDF version 1.6 (*Adobe*, *2004*)), published in 2004, was the first to support so-called "3D Artwork" as an embedded multimedia feature. In January 2005, the Acrobat 7 product family provided the first implementation of tools for creating and displaying these 3D models (*Adobe*, *2005*).

The latest version (PDF version 1.7 (*Adobe*, *2008a*)) supports three types of geometry (meshes, polylines and point clouds), textures, animations, 15 render modes, 11 lighting schemes and several other features. The only 3D file format that is supported by the ISO standard (*ISO*, *2008*) is Universal 3D (U3D, see section below). Support for another 3D format (Product Representation Compact, PRC) has been added by Adobe (*Adobe*, *2008b*) and has been proposed to be integrated into the replacement Norm ISO 32000-2 (PDF 2.0). However, this new standard is currently only available as draft version (*ISO*, *2014*) and has not yet been adopted.

Although the first application in scientific context was proposed in November 2005 (*Zlatanova & Verbree*, *2005*) and thus quite soon after this new technology was available, it took three more years before the first applications were really demonstrated in scholarly articles (*Ruthensteiner & Heß*, *2008*; *Kumar et al.*, *2008*; *Barnes & Fluke*, *2008* ). Since then, the number of publications that apply PDF 3D technology either in theory or in practice has increased almost every year (Table 1). The most sophisticated implementation so far is the reporting of planning results for liver surgery where the PDF roots are hidden behind a user interface which emulates a stand-alone software application (*Newe, Becker & Schenk*, *2014*).

## The Universal 3D (U3D) file format

As outlined above, the U3D file format is the only 3D format that is supported by the current ISO specification of PDF. Initially designed as an exchange format for Computer Aided Construction (CAD), it was later standardized by ECMA International (formerly known as European Computer Manufacturers Association, ECMA) as ECMA-363 (Universal 3D File Format). The latest version is the 4th edition from June 2007 (*ECMA*, *2007*).

U3D is a binary file format that comprises all information to describe a 3D scene graph. A U3D scene consists of an arbitrary number of objects that can be sorted in an object tree. The geometry of each object can be defined as a triangular mesh, a set of lines or a set of points. A proprietary bit encoding algorithm allows for a highly compressed storage of the geometry data. A number of additional features and entities (textures, lighting, views, animations) can be defined; details are described in previously published articles (*Newe & Ganslandt*, *2013*).

The scholarly publishing company Elsevier invites authors to supplement their articles with 3D models in U3D format (*Elsevier*, *2015*) and many 3D software tools provide the possibility to export in U3D format. However, most of them are commercial software, but open source solutions like MeshLab (http://meshlab.sourceforge.net/) are available as well.

## Creating 3D model files and PDF documents

Although many tools and libraries are available that support the creation of 3D model files and of final PDF documents, the whole process is still cumbersome. The problems are manifold: some tools require programming skills; some do not support features those are of interest for scientific 3D data (like polylines (*Newe*, *2015*) and point clouds *Barnes & Fluke*, *2008*). Operating system platform support is another issue, as well as royalty-free use.

As regards the creation of the 3D model files, most of these problems have been addressed in a previous article (*Newe*, *2015*). The main problem, however, remains the creation of the final PDFs. Specifying the content and (in particular) the layout of a document can be a complex task and is usually the domain of highly specialized word processor software. Figures and supplements for scholarly publications, on the other hand, usually have a specific layout where only the contents of (a limited number of) pre-defined elements vary. There are at least three common elements for a scientific figure: the figure itself, a short caption text and a longer descriptive text. If the figure is intended to be provided as supplemental information file instead of being integrated into the main article text, some additional information is necessary as well: at least a general headline and an optional reference to the main article should be provided. If the document content is modularized to these five key elements (Fig. 1), the creation of the PDF itself becomes a rather simple task, because the layout can be pre-defined.

One last difficulty arises from a peculiarity of interactive 3D figures in PDF: the number viewing options (e.g., camera angle, zoom, lighting…) is nearly unlimited. Although such a figure is intended to provide all these options, an author usually wants to define an initial view at the objects, if only to simply ensure that all objects are visible. No freely available tool for PDF creation currently provides a feature to pre-define such a view. The movie15 package for LaTeX (*Grahn*, *2005*) provides a mechanism do determine the view parameters, but that requires the generation of intermediate PDFs.

Finally it must be mentioned that many previously published 3D models are very large—sometimes up to nearly 100 megabytes (*Krause et al.*, *2014*). In most cases, this size could (and should) be reduced significantly, because the density of polygon meshes does usually not need to be very high for illustrative purposes.

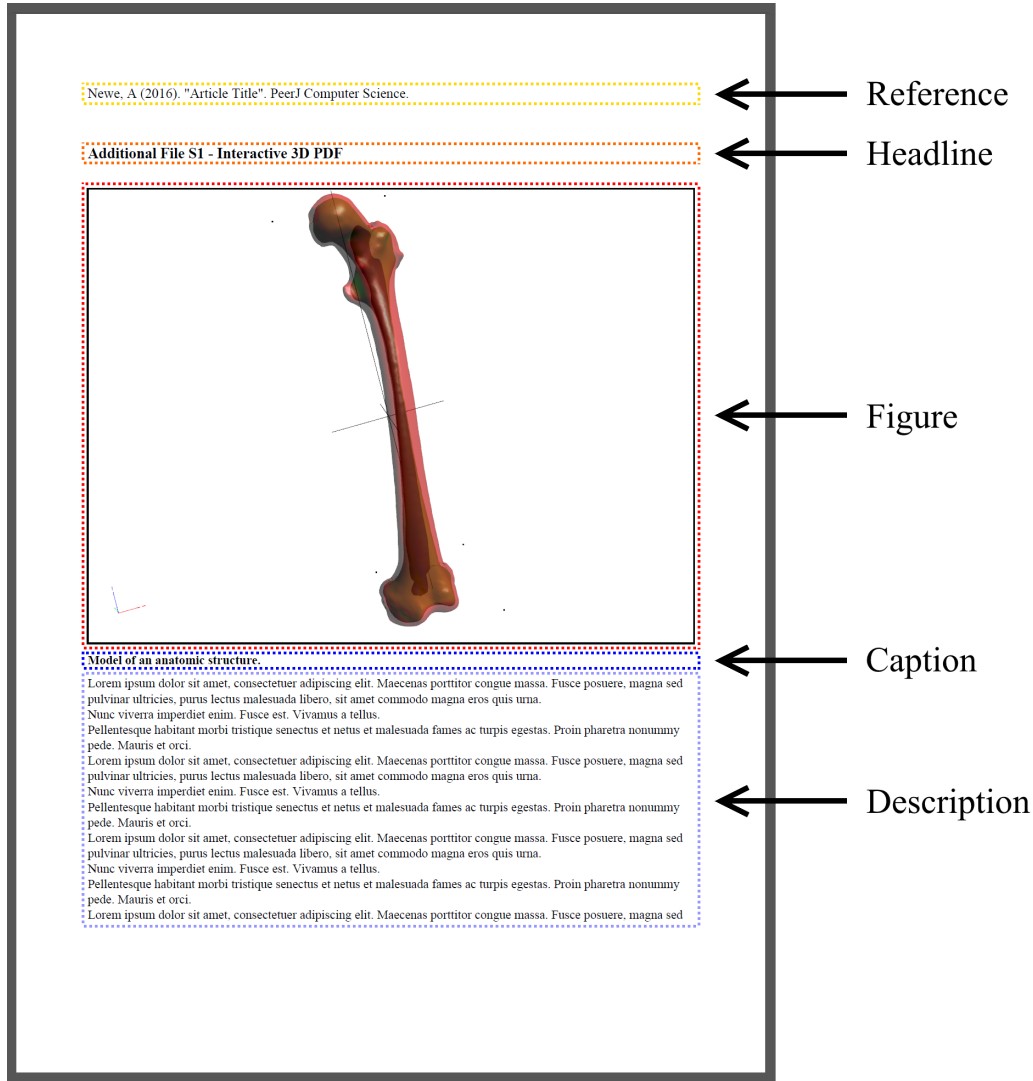

Newe, A (2016). "Article Title". PeerJ Computer Science. ← Reference

Additional File S1 - Interactive 3D PDF ← Headline

← Figure

Model of an anatomic structure. ← Caption

Lorem ipsum dolor sit amet, consectetuer adipiscing elit. Maecenas porttitor congue massa. Fusce posuere, magna sed pulvinar ultricies, purus lectus malesuada libero, sit amet commodo magna eros quis urna.
Nunc viverra imperdiet enim. Fusce est. Vivamus a tellus.
Pellentesque habitant morbi tristique senectus et netus et malesuada fames ac turpis egestas. Proin pharetra nonummy pede. Mauris et orci.
Lorem ipsum dolor sit amet, consectetuer adipiscing elit. Maecenas porttitor congue massa. Fusce posuere, magna sed pulvinar ultricies, purus lectus malesuada libero, sit amet commodo magna eros quis urna.
Nunc viverra imperdiet enim. Fusce est. Vivamus a tellus.
Pellentesque habitant morbi tristique senectus et netus et malesuada fames ac turpis egestas. Proin pharetra nonummy pede. Mauris et orci.
Lorem ipsum dolor sit amet, consectetuer adipiscing elit. Maecenas porttitor congue massa. Fusce posuere, magna sed pulvinar ultricies, purus lectus malesuada libero, sit amet commodo magna eros quis urna.
Nunc viverra imperdiet enim. Fusce est. Vivamus a tellus.
Pellentesque habitant morbi tristique senectus et netus et malesuada fames ac turpis egestas. Proin pharetra nonummy pede. Mauris et orci.
Lorem ipsum dolor sit amet, consectetuer adipiscing elit. Maecenas porttitor congue massa. Fusce posuere, magna sed ← Description

**Figure 1** **General layout of a scholarly figure if provided as supplemental material**

## MeVisLab

MeVisLab is a framework for image processing and an environment for visual development, published by MeVis Medical Solutions AG and Fraunhofer MEVIS in Bremen, Germany. It is available via download (http://www.mevislab.de/download/) for all major platforms (Microsoft Windows, Mac OS and Linux) and has a licensing option which is free for use in non-commercial organizations and research ("MeVisLab SDK Unregistered" license, http://www.mevislab.de/mevislab/versions-and-licensing/). Beside the development features, MeVisLab can be used as a framework for creating sophisticated applications with graphical user interfaces that hide the underlying platform and that can simply be used without any programming knowledge (*Koenig et al.*, *2006*; *Heckel, Schwier & Peitgen*, *2009*; *Ritter et al.*, *2011*). MeVisLab has been evaluated as a very good platform for creating application prototypes (*Bitter et al.*, *2007*), is very

well documented (http://www.mevislab.de/developer/documentation/) and supported by an active online community (http://www.mevislab.de/developer/community/; https://github.com/MeVisLab/communitymodules/tree/master/Community).

All algorithms and functions included into MeVisLab are represented and accessed by "modules," which can be arranged and connected to image processing networks or data processing networks on a graphical user interface (GUI) following the visual data-flow development paradigm. By means of so-called "macro modules," these networks can then be converted with little effort into complete applications with an own GUI.

## METHODS

### Elicitation of requirements

As described above, the generation of the necessary 3D model data and particularly of the final PDF is still subject to a number of difficulties. Therefore, the first step was the creation of a list of requirement specifications with the aim to create a tool that overcomes these known drawbacks.

Two requirements have been identified to be the most important ones: (1) the demand for a tool that creates "ready-to-publish" PDF documents without the need for commercial software and (2) the integration of all necessary steps into a single and easy-to-use interface. Beside these two main requirements, a number of additional requirements have then been identified as well. See Table 2 for a full list of all requirements that were the basis for the following development.

### Creation of an "App" for MeVisLab

MeVisLab-based solutions presented in previous work (*Newe & Ganslandt*, *2013*; *Newe*, *2015* ) already provide the possibility to create U3D files without requiring programming skills and without the need for an intensive training. However, they still needed some basic training as regards assembling the necessary processing chains in MeVisLab. Furthermore, the creation of the final PDF was not possible so far.

Therefore, a new macro module was created for MeVisLab. A macro module encapsulates complex processing networks and can provide an integrated user interface. In this way, the internal processes can be hidden away from the user, who can focus on a streamlined workflow instead. Designed in an appropriate way, a macro module can also be considered as an "app" inside of MeVisLab.

In order to provide the necessary functionality, some auxiliary tool modules (e.g., for the creation of the actual PDF file) needed to be developed as well. Along with the modules for U3D export mentioned above, these auxiliary tool modules were integrated into the internal processing network of the main "app" macro. The technical details of these internal modules are not within the scope of this article. However, the source code is available and interested readers are free to explore the code and to use it for own projects.

The user interface of the app was designed in a way that it guides novice users step-by-step without treating experienced users too condescendingly, though. Finally, a comprehensive documentation including an example project, a wizard for creating tailored PDF modules and a verbose help text was set up.

**Table 2  Requirements for the development of the software tool.** The two main requirements are highlighted in bold font.

| ID | Requirement specification |
| --- | --- |
| **R1** | **The software *shall* create ready-to-publish PDF documents with embedded 3D models.** |
| R1.1 | The software *shall* offer an option to specify the activation mode and the deactivation mode for the 3D models. |
| **R2** | **The software *shall* provide an integrated, single-window user interface that comprises all necessary steps.** |
| R3 | The software *shall* be executable under Windows, MacOS and at least one Linux distribution. |
| R4 | The software *shall* be executable without the need to purchase a commerical license. |
| R5 | The software *shall* create 3D model files in U3D format. |
| R5.1 | The software *shall* create view definitions for the 3D model. |
| R5.2 | The software *shall* create poster images for the PDF document. |
| R6 | The software *shall* import mesh geometry from files in OBJ, STL and PLY format. |
| R6.1 | The software *should* import mesh geometry from other file formats as well. |
| R6.2 | The software *shall* offer an option to reduce the number of triangles of imported meshes. |
| R6.3 | The software *shall* offer an option to specify the U3D object name and the color of imported meshes. |
| R7 | The software *shall* import line set geometry from files in text format. |
| R7.1 | The software *shall* offer an option to specify the U3D object name and the color of imported line sets. |
| R8 | The software *shall* import point set geometry from files in text format. |
| R8.1 | The software *shall* offer an option to specify the U3D object name of imported point sets. |

## Deployment of core functionality

For the creation of the actual PDF files, version 2.2.0 of the cross-platform, open source library libHaru (http://libharu.org/) was selected, slightly modified and integrated as third-party contribution into MeVisLab.

Next, the application programming interface (API) of libHaru was wrapped into an abstract base module for MeVisLab in order to provide an easy access to all functions of the library and in order to hide away standard tasks like creating a document or releasing memory. A large number of convenience functions were added to this base module and an exemplary MeVisLab project was set up in order to demonstrate how to use the base module for tailored applications. This base module also served as basis for the PDF creation of the app macro described above. Finally, a project wizard was integrated into the MeVisLab GUI.

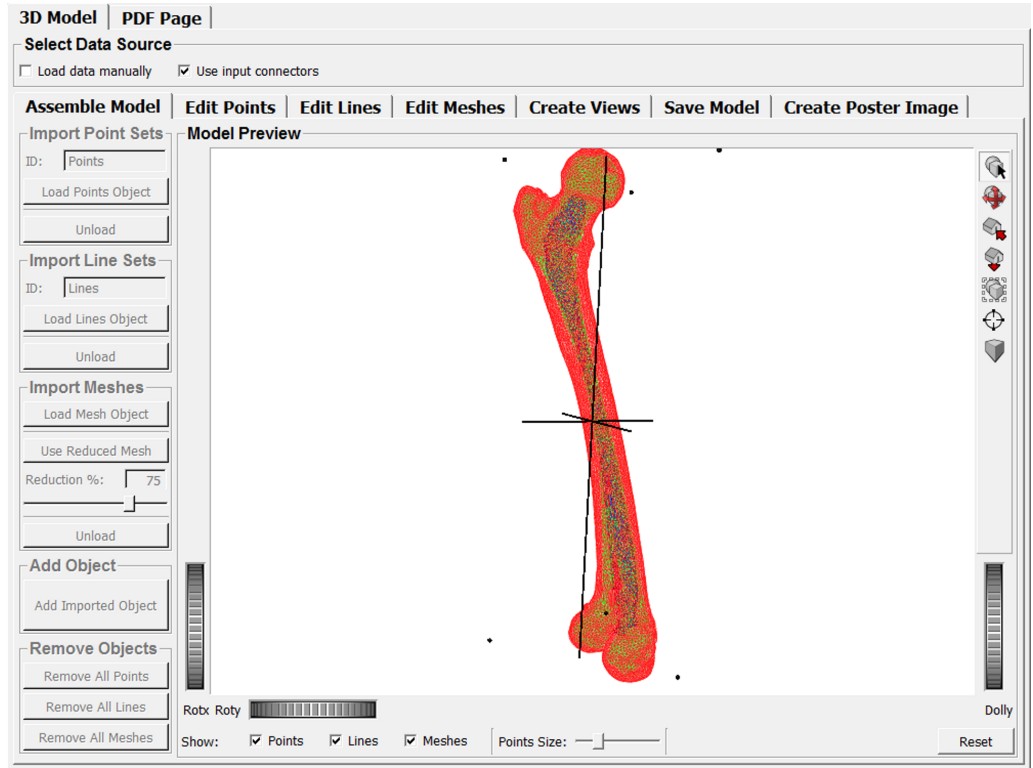

**Figure 2** **User interface of the app.** The user interface comprises all necessary steps for the creation of 3D model files and PDF files. It is arranged in tabs for each step.

## RESULTS

### The "Scientific3DFigurePDFApp" module

The new macro module "Scientific3DFigurePDFApp" for MeVisLab provides an integrated user interface for all steps that are necessary for the creation of U3D model files and for the creation of the final PDF documents with embedded 3D models. The model editor part produces U3D model files of geometry data that are compatible with version 4 of the ECMA-363 standard and poster images in Portable Network Graphics (PNG) format. The PDF editor part produces PDF documents that are compliant with PDF version 1.7 (ISO 32000-1:2008). An example PDF is available as File S1.

The user interface is arranged in tabs, whereas each tab comprises all functions for one step of the workflow. By processing the tabs consecutively, the user can assemble and modify 3D models, save them in U3D format, create views and poster images for the PDF document, and finally create the PDF itself step by step (Fig. 2).

The raw model data can be collected in two ways: either by feeding it to the input connectors or by assembling it by means of the built-in assistant. The former option is intended for experienced MeVisLab users that want to attach the module at the end of a processing chain. The latter option addresses users that simply want to apply the app for converting existing 3D models and for creating an interactive figure for scholar publishing.

**Table 3** List of supported 3D formats for importing geometry data (textures, animations and other features are not supported).

| File format | Typical file extension(s) |
| --- | --- |
| Stereolithography | *.stl |
| Stanford Polygon Library | *.ply |
| Wavefront Object | *.obj |
| Object File Format | *.off |
| Blender | *.blend |
| Raw Triangles | *.raw |
| Raw Point Clouds | *.csv; *.txt |
| Raw Line Sets | *.csv; *.txt |
| 3D GameStudio | *.mdl; *.hmp |
| 3D Studio Max | *.3ds; *.ase |
| AC3D | *.ac |
| AutoCAD/Autodesk | *.dxf |
| Biovision BVH | *.bvh |
| CharacterStudio Motion | *.csm |
| Collada | *.dae; *.xml |
| DirectX X | *.x |
| Doom 3 | *.md5mesh; *.md5anim; *.md5camera |
| Irrlicht | *.irrmesh; *.irr; *.xml |
| LightWave | *.lwo; *.lws |
| Milkshape 3D | *.ms3d |
| Modo Model | *.lxo |
| Neutral File Format | *.nff |
| Ogre | *.mesh.xml; *.skeleton.xml; *.material |
| Quake I, Quake II, Quake III | *.mdl; *.md2; *.md3; *.pk3 |
| Quick3D | *.q3o; *q3s |
| RtCW | *.mdc |
| Sense8 WorldToolkit | *.nff |
| Terragen Terrain | *.ter |
| TrueSpace | *.cob; *.scn |
| Valve Model | *.smd; *.vta |
| XGL | *.xgl; *.zgl |

The software allows for importing the geometry data of 39 different 3D formats, including point clouds and line sets from files in character-separated value (CSV) format (see Table 3 for a full list). The import of textures and animations is not supported.

Objects from different sources can be combined and their U3D properties (colour, name, position in the object tree) can be specified. The density of imported meshes can be adjusted interactively and multiple views (i.e., the specification of camera, lighting and render mode) can be pre-defined interactively as well. Finally, it is also possible to create a poster image which can replace an inactive 3D model in the PDF document if the model itself is disabled or if it cannot be displayed for some reason (e.g., because the reading software does not provide the necessary features).

**Table 4  List of features.**

| Category | Features |
| --- | --- |
| Data import | Import external data, import MeVisLab data, import point clouds, import line sets, import meshes from 37 file formats, adjust mesh density, preview import |
| Point cloud editing | Specify point cloud name, specify position in model tree, preview settings |
| Line set editing | Specify line set name, specify position in model tree, specify colour, preview settings |
| Mesh editing | Specify mesh name, specify position in model tree, specify colour, specify opacity, preview settings |
| View specification | Specify view name, specify background colour, specify lighting scheme, specify render mode, preview settings, specify multiple views |
| U3D creation | Store model in U3D format, preview scene |
| Poster image creation | Store poster in PNG format, preview scene, specify superimposed text |
| PDF creation | Store document in PDF (v1.7) format, specify header citation text, specify header headline text, specify U3D file, specify poster file, specify model activation mode, specify model deactivation mode, specify toolbar enabling, specify navigation bar enabling, specify animation start mode, specify caption, specify description text |

All functions are explained in detail in a comprehensive documentation which can be accessed directly inside MeVisLab. A stand-alone copy of the documentation is available as File S2. In order to use the app, it simply needs to be instantiated (via the MeVisLab menu: Modules → PDF → Apps → Scientific3DFigurePDFApp). A full feature list is available in Table 4.

### Additional features for tailored PDF creation

The abstract module which wraps the API of the PDF library libHaru into a MeVisLab module was made public ("PDFGenerator" module) and can be used for the development of tailored MeVisLab modules. In order to facilitate the re-use of this abstract base module, an exemplary project was set up (/Projects/PDFExamples/SavePDFTemplate). This project demonstrates how to derive a customized module from the PDFGenerator base module and how to specify the content of the PDF file that will be created by means of the new module. The template code is verbosely annotated and includes examples for setting PDF properties (e.g., meta data, page size, encryption) as well as the document content (including text, images, graphics and 3D models). The output of the SavePDFTemplate module is illustrated in Fig. 3.

Finally, a project wizard was integrated into the MeVisLab GUI. It can be accessed via the MeVisLab menu: File → Run Project Wizard…→ PDF Module. The wizard consists of three steps (Fig. 4) whereof the second step offers the possibility to include demo code which produces the same PDF file as the SavePDFTemplate module described above. The general usage of project wizards in MeVisLab is explained in chapter 23 of the MeVisLab

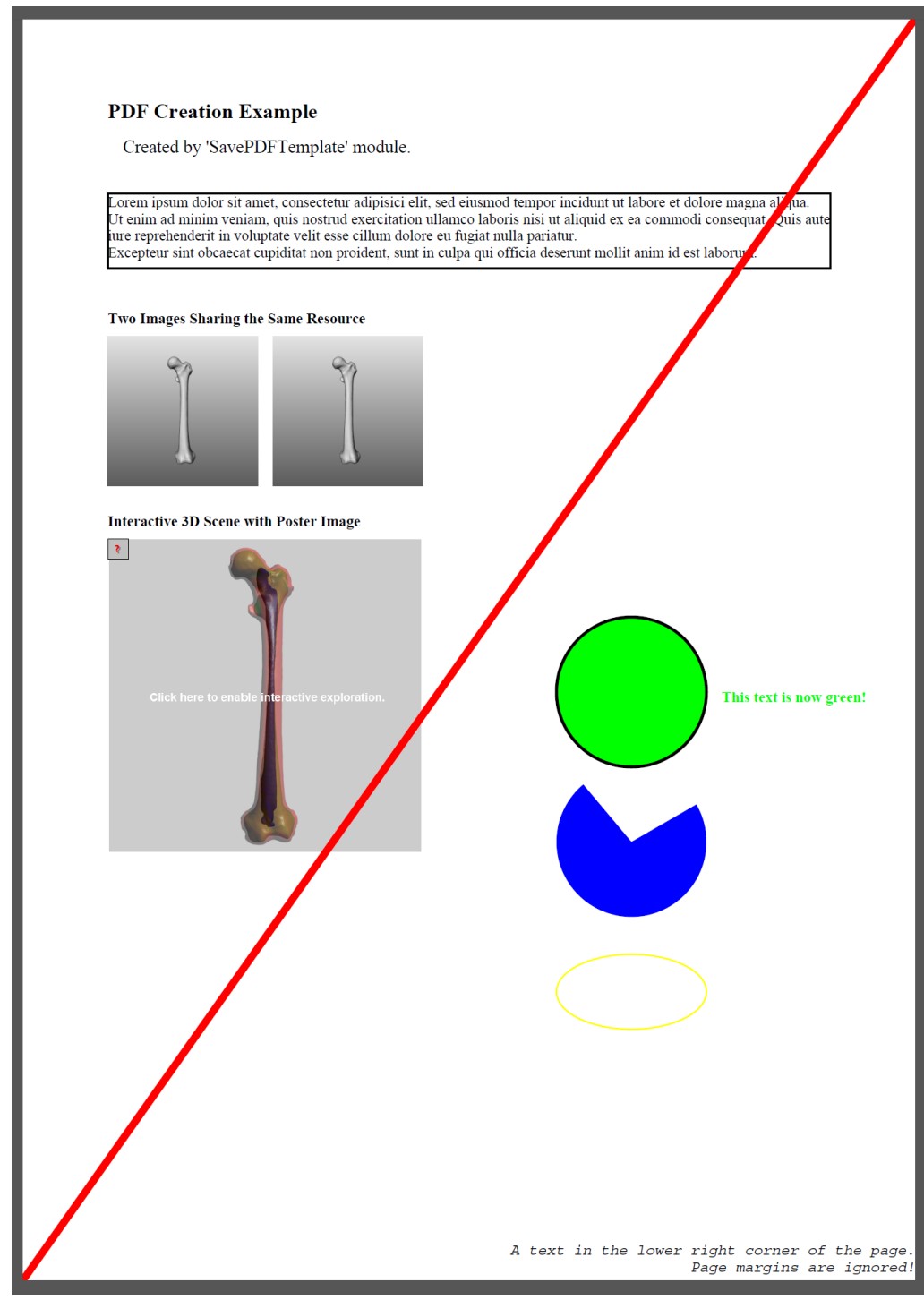

**Figure 3** **Output of the SavePDFTemplate module.**

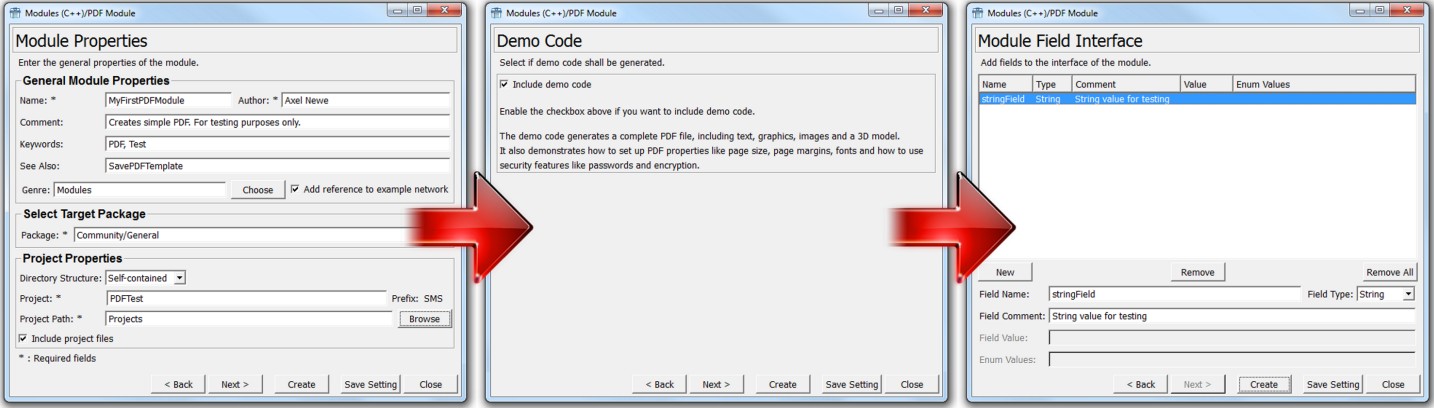

**Figure 4** Project wizard for creating customized PDF modules.

manual (menu: Help → Browse Help Pages → Using MeVisLab → MeVisLab Reference Manual).

## Availability

The whole PDF project for MeVisLab (which includes the Scientific3DFigurePDFApp, the PDFGenerator base module, the SavePDFTemplate project, the project wizard and all source code files) is available for Microsoft Windows, MacOS and Linux (tested with Ubuntu 14.04.2). It requires MeVisLab 2.8 or a later version (http://www.mevislab.de/download/). The Windows version of MeVisLab is available for four compiler versions. This is relevant only if the source code is intended to be compiled manually (see below).

There are two approaches to add the app and the other elements to an existing MeVisLab installation: add-on installers and the online repository of the MeVisLab community sources.

Installers are self-contained, executable archives that automatically add all necessary files to an existing MeVisLab installation. The target groups for these installers are MeVisLab newcomers and pure users that want to use the Scientific3DFigurePDFApp out-of-the-box. The current version of the installers for all operating systems and all 64-Bit compiler versions can be downloaded from the research data repository Zenodo (10.5281/zenodo.48758). Installers for the previous version of MeVisLab (2.7.1) are available as well (10.5281/zenodo.47491), but this version will not be supported in the future. Updates will be made available via Zenodo as well. A dedicated Zenodo Community Collection named "Three-dimensional Portable Document Format (3D PDF) in Science" has been set up for this purpose (https://zenodo.org/collection/user-3d-pdf).

All those who are interested in being able to always use the latest version should connect their MeVisLab installation with the community sources which are hosted at GitHub (https://github.com/MeVisLab/communitymodules/tree/master/Community). This approach, however, requires compiling the source code and is intended only for experienced users or for users that are willing to become acquainted with MeVisLab. Note

that there are multiple versions available for Windows, depending on the compiler that is intended to be used.

## DISCUSSION

### A toolbox for the creation of 3D PDFs

The utilization of 3D PDF technology for scholarly publishing has been revealed and proven both useful and necessary by several authors in the past years. The mainstream application of 3D PDF in science, however, is yet to come.

One reason might be the difficult process that has so far been necessary to create appropriate data and relevant electronic documents. This article presents an all-in-one solution for the creation of such files which requires no extraordinary skills. It can be used by low-end users as an out-of-the-box tool as well as a basis for sophisticated tailored solutions for high-end users.

Many typical problems as regards the creation of 3D model files have been addressed and solved. All steps of the workflow are integrated seamlessly. The software is available for all OS platforms and can import and process objects from many popular 3D formats, including polylines and point clouds (Table 3). The density of imported meshes can be adjusted interactively which enables the user to find the best balance between the desired level of detail and the file size.

The main contribution, however, is the possibility to create ready-to-publish PDF documents with a minimum of steps. This approach was proposed to be the ideal solution by *Kumar et al. (2010)*. To best knowledge, this is the first time that such an integrated and comprehensive solution is made available for the scientific community.

### Applications

The areas of application (see an example in File S1) are manifold and not limited to a specific scientific discipline. On the contrary: every field of research that produces three-dimensional data can and should harness this technology in order to get the best out of that data.

One (arbitrary) example for the possible use of mesh models from the recent literature is 3D ultrasound. *Dahdouh et al.* (*2015*) recently published about the results of segmentation of obstetric 3D ultrasound images. That article contains several figures that project three-dimensional models on the available two-dimensional surface. A presentation in native 3D would have enabled the reader to interactively explore the whole models instead of just one pre-defined snapshot. Another example is the visualization of molecular structures as demonstrated by *Kumar et al. (2008)*.

Polylines can be used to illustrate nervous fibre tracking. *Mitter et al., (2015)* used 2D projections of association fibres in the foetal brain to visualize their results. A real 3D visualization would have been very helpful in this case as well: while some basic knowledge about a depicted object helps to understand 2D projections of 3D structures, the possibility to preserve at least a little depth perception decreases with an increasing level of abstraction (mesh objects vs. polylines).

This particularly applies to point clouds which can be observed, for example, in an article by *Qin et al. (2015)*: although these authors added three-dimensional axes to their figure (no. 6) it is still hard to get an impression of depth and therefore of the real position of the points in 3D space.

## Limitations

Although the presented software pulls down the major thresholds that impede the creation of interactive figures for scholarly publishing, some limitations still need to be considered.

A general concern is the suitability of PDF as a means to visualize and to exchange 3D models. PDF and U3D (or PRC) do not support all features that other modern 3D formats provide and that would be of interest for the scientific community (e.g., volumetric models). On the other hand, PDF is commonly accepted and de-facto the only file format that is used for the electronic exchange of scholarly articles. Therefore, PDF may not be the perfect solution, but it is the best solution that is currently available.

The presented software requires MeVisLab as background framework and the installation of MeVisLab requires a medium-sized download of about 1 GB (depending on the operating system), which could be considered rather large for a PDF creator. On the other hand, MeVisLab integrates a large library for the processing and the visualization of (biomedical) image data. Furthermore, other frameworks (like MeshLab) do not provide all necessary features (e.g., polylines or point clouds) and therefore were not considered to meet basic requirements for the development of the software tool.

The import of 3D models is based on the Open Asset Import Library (http://www.assimp.org/) which does not support all features of all 3D formats. For example, textures and animations cannot be imported and should thus not be embedded into a model file that is intended to be imported. However, although the model-editor part of the presented software does not support textures (or animations), the PDF-creator part can still be used to produce scientific PDFs with textured or animated models, if the necessary U3D files have been created with external and more specialized software. In this use case, the Scientific3DFigurePDFApp does not integrate all necessary steps, but it still remains a "few-clicks" alternative for the creation of interactive PDF supplements for scientific publications and it still obviates the need for a commercial solution.

Finally, very large model files should be avoided. If a large model fails to import, it should be separated into several sub-models. A mesh reduction can be applied after the import, but a previously reduced mesh speeds up the import process.

## Suitable reading software

The Adobe Reader (http://get.adobe.com/reader/otherversions/) is available free of charge for all major operating systems (MS Windows, Mac OS, Linux). It is currently the only software that can be used to display embedded 3D models and to let the user interact with them (zooming, panning, rotating, selection of components). However, even the Adobe Reader does not support all U3D features (*Adobe, 2007*), e.g., Glyphs and View Nodes. Furthermore, a rendering flaw has been observed on low-end graphic boards in MacOS hardware (Fig. 5). Adobe Reader for MacOS does not render transparent surfaces

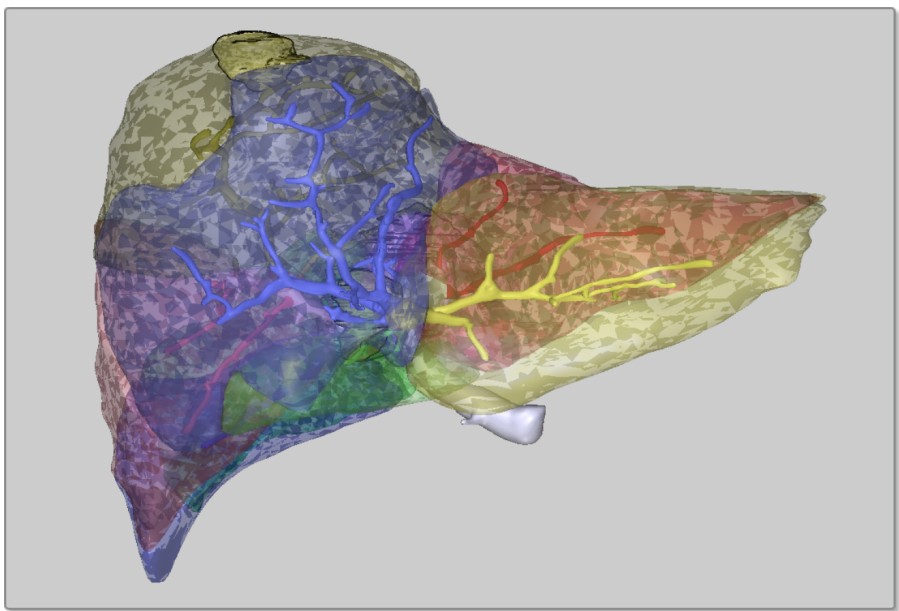

**Figure 5** **Rendering artifacts.** These tessellation artifacts have been observed on MacOS systems with low-end graphic hardware.

superimposed upon each other correctly: instead, a strong tessellation effect is visible. This may also occur on other platforms but has not been reported yet. Since this is an issue with the rendering engine of Adobe Reader, there is currently no other solution than using a different render mode (e.g., one of the wireframe modes) or different hardware.

Experience shows that many users do not expect a PDF document to be interactive. Therefore, possible consumers should be notified that it is possible to interact with the document and they should also be notified that the original Adobe Reader is required for this. Although poster images are a workaround to avoid free areas in PDF readers that are not capable of rendering 3D scenes, missing 3D features of a certain reader could be confusing for a user.

## A basis for own modules

As pointed out in previous work (*Newe, 2015*), the authoring of a PDF document is usually a complex task and thus in most cases it cannot be avoided to separate the generation of 3D model data from the actual PDF authoring. Although the software tool presented in this article mitigates this general problem by integrating model generation and PDF creation, it is still limited to a certain use case and a pre-defined PDF layout.

However, the API of the core PDF functionality is public and designed in a way that facilitates the creation of own PDF export modules. The large number of convenience functions for the abstract base module (PDFGenerator) facilitates the creation of derived modules. These functions massively lighten the programmer's workload by providing a simple access to routine tasks like writing text at a defined position or like embedding a 3D model which would normally require a whole series of API calls. Finally, the built-in

wizard generates all necessary project files and source code files to create a fully functional module barebone which only needs to be outfitted with the desired functionality.

### Outlook

Although this article represents an important milestone, the development of the PDF project for MeVisLab is ongoing. Future goals are the integration of virtual volume rendering (*Barnes et al.*, *2013*), animations (*Van de Kamp et al.*, *2014*) and the parsing of U3D files that have been created with external software. The progress can be tracked via GitHub (https://github.com/MeVisLab/communitymodules/tree/master/Community) and updates to the binary files will be published regularly.

## CONCLUSION

Three-dimensional data is produced at an increasing rate throughout all scientific disciplines. The Portable Document Format is a well-suited medium for the visualization and the publication of this kind of data. With the software presented in this article, the complete workflow for generating 3D model files and 3D PDF documents for scholarly publications can be processed in a consolidated working environment, free of license costs and with all major operating systems. The software addresses novices as well as experienced users: on the one hand, it provides an out-of-the-box solution that can be used like a stand-alone application, and on the other hand, all sources and APIs are freely available for specifically tailored extensions.

### List of abbreviations

| | |
|---|---|
| **2D** | Two-dimensional |
| **3D** | Three-dimensional |
| **PDF** | Portable Document Format |
| **ISO** | International Organization for Standardization |
| **U3D** | Universal 3D |
| **PRC** | Product Representation Compact |
| **CAD** | Computer Aided Construction |
| **ECMA** | European Computer Manufacturers Association |
| **GUI** | Graphical User Interface |
| **API** | Application Programming Interface |
| **PNG** | Portable Network Graphics |
| **CSV** | Character-separated Value, Comma-separated Value |

## ACKNOWLEDGEMENTS

Thanks to Dr. Hans Meine, Fraunhofer MEVIS, Bremen, Germany, for compiling the code for MacOS and Linux and for building installers. Thanks to Dr. Thomas van de Kamp, Karlsruhe Institute of Technology, Karlsruhe, Germany, for providing test data.

### Funding

The author received no funding for this work.

### Competing Interests

The author declares there is no competing interests.

### Author Contributions

- Axel Newe conceived and designed the experiments, contributed reagents/materials/analysis tools, wrote the paper, prepared figures and/or tables, performed the computation work, reviewed drafts of the paper, implemented the software.

### Data Availability

https://github.com/MeVisLab/communitymodules/tree/master/Community/General;

10.5281/zenodo.48758;

10.5281/zenodo.47491.

### Supplemental Information

Supplemental information for this article can be found online at http://dx.doi.org/10.7717/peerj-cs.64#supplemental-information.

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
