# Peer review of "Enriching scientific publications with interactive 3D PDF: an integrated toolbox for creating ready-to-publish figures"

_PeerJ Computer Science, doi:10.7717/peerj-cs.64_

## Round 0.1 · original submission · Major Revisions

The manuscript has been reviewed by our advisers. Their reports appear below.

In view of these comments we cannot accept the paper for publication in its current form. Should you be prepared to incorporate major revisions, please carefully consider the reviewers' comments and submit a list of detailed responses to the comments. Your revised manuscript will be reconsidered for publication in PeerJ.

·

Basic reporting

Article generally well written. I suggest table 2 could become main text and some of the reasons behind those desirable features discussed/justified.

Experimental design

Experimental design not relevant.

Validity of the findings

Agree with the author that 3D pdf has been a difficult to author and is thus an impediment to uptake. The problem is that 3D PDF is not new and things have not significantly improved over the last 5 years. I struggle to agree that the proposed solution will significantly improve the situation.

To the question of 3D PDF being a suitable format. The author says “… images and multimedia elements like audio, movies or 3D models”. Lots of 3D data structures are not supported and are key for visualisation, e.g.: volumetric. But even for geometry, basic primates like spheres and cylinders are not defined except as very inefficient “triangle soup” mesh representations.

The author says in reference to model sizes “In most cases, this size could (and should) be reduced significantly, because the density of polygon meshes does usually not need to be very high for illustrative purposes.” If one can zoom into models then why might a high resolution mesh not be desirable? Data is becoming increasingly detailed and high resolution so I do not accept that model sizes are necessarily low. I would further argue that simple 3D models are more likely to be fully conveyed in 2D, it is exactly the complicated models that require a interactive navigable experience to assist with understanding. It should also be noted that the 3D PDF does not support what one might be considered standard features like progressive mesh rendering or even multiple levels of detail.

With respect to the import format list it is indeed impressive, but sorry but I do not accept these formats in their full specification are supported. To test I wasn't unduly negative I tried to import a textured 3D mesh as an OBJ file and it failed, didn't have time to identify why but suspect it was due to multiple large texture files. It was a very standard OBJ file that I can open in every other software package I have tried.

Additional comments

I don't wish to sound too harsh, I like many were initially intrigued by the possibilities 3D PDF may offer. The issues I have with 3D PDF are
1. Lack of viewer support outside one companies product.
2. Lack of diversity of representation within the format.
3. Lack of tools for creating 3D PDF.
There seems to be a lack of support for any pdf viewer other than Adobe product. Where is the “export to pdf” in 3D modelling/editing packages? No support from the leading work processing solution. These tell me that widespread support from the industry is not growing.

See two attachments, "PoorShading" shows the coarse model supplied as an example, aren't vertex normals supported? "CorruptDrawing", Adobe Acrobat viewer didn't seem to handle OS level screen zooming. Not a complaint of this paper but further evidence of an immaturity of support. I also struggled to find support on mobile devices.

Unfortunately the article has not allayed my concerns. The solution presented is semi-commercial and a 4+GB download, seems like overkill just to create 3D PDFs.

I might agree with the aims of this paper say 5 years ago, as a proposed future opportunity to publish 3D data with documents, but today it would seem the promise as a suitable medium has not been realised. Similar to the push for VRML in the late 80’s, the reality has been quite different. There are many reasons for this, one is the complexity and diversity needed for 3D data representation requirements, another key factor is the lack of cross platform and software support. In summary, I don’t believe the case has been made that 3D PDF has a future for scientific data visualisation and that it supports the type of 3D data representations researchers require.

Final comment, if the author resubmits then I strongly suggest creating an example that makes a strong case.

Reviewer 2 ·

Basic reporting

The paper is well written and merits publication in PeerJ.

Experimental design

The work reported in this manuscript seems to have been carried out with care.

Validity of the findings

I think that the software toolbox for generating 3D PDF documents could become a very useful format in scientific publications.

Additional comments

It would be useful to comment on how this toolbox could be used to visualize 3D structures of molecules and proteins.

Reviewer 3 ·

Basic reporting

Comments (PeerJ) 9236-v0
Title: Enriching scientific publications with interactive 3D PDF figures: A complete toolbox

Author: Axel Newe

This manuscript presents a developed tool box for 3D-pdf file presentation of scientific visualization through the Adobe software. Development of such interactive 3D visualization with easy access is useful for readers to gain a thorough understanding of the problems presented. It is particularly useful for medicinal and biological applications such as protein docking etc. From this point of view, this article addresses a useful and interesting issue.

However, I do not believe that this manuscript is publishable at its present form as the 3D-pdf based on in complete literature review and therefore, this manuscript missed significant milestone work in this direction, which may shake the foundation of this study and make it redundant.

First of all, the manuscript missed the significant development in interactive 3d-pdf based on the Adobe software/tool box in 2009, in which the authors detailed the development of the 3D-pdf to show the 3D structures of drugs using embedded 3d-pdf technique they developed.

1. Lalitah Selvam, Vladislav Vasilyev and Feng Wang, Methylation of zebularine: a quantum mechanical study incorporating interactive 3D PDF graphs, Journal of Physical Chemistry B 113, 11496-11504(2009).
2. L. Selvam, F. F. Chen and F. Wang, Methylation of zebularine investigated using density functional theory calculations, Journal of Computational Chemistry, 32(10)(2011)2077–2083.
3. A. P. Wickrama Arachchilage, F. Wang, V. Feyer, O. Plekan, and K. C. Prince, “Photoelectron spectra and structures of three cyclic dipeptides: PhePhe, TyrPro and HisGly”, Journal of Chemical Physics, 136, 124301 (2012) (8 pages).

The JCP article was in fact selected as the cover page for its 3D structure of the issue. None of the articles are referred in this manuscript. As the author missed these articles using very similar technique, unless such related articles are properly referred and the technique of the manuscript is justified, I do not think this article of a very similar technique is worthy publishing.

Experimental design

NA

Validity of the findings

NA

Additional comments

The author need a thorough literature review and developing something existing is extremely time consuming.

Reviewer 4 ·

Basic reporting

English is good.
Introduction and background are well written and clearly position the paper in the relevant field.
Figure are relevant, looks good.

Experimental design

This is a paper about software.
I am not sure how it fits to the requirements.
However I presume the design of the program is OK - I have no way to assess it.

Validity of the findings

The example in the Supplementary materials works really well on my computer: Dell notebook about 4 years old.
I think it demonstrates the validity of the project.

Additional comments

I did not install the program and my review is really deficient due to this.
There are two reasons why I did not install the program:
1.
The download page for MeVisLab
http://www.mevislab.de/download/

indicates the size of the installation
MeVisLabSDK2.8_vc12-64.exe (1154 MB)

This is more than 1 Gigabytes.
It is really too large for me.

2.
I was not sure that the above download is what I need.
The download page talks about Microsoft compilers.
Do I need to compile the installation with them?
The size if the installation is too large to just try it.

---

## Round 0.2 · Minor Revisions

We have received two further reviews from our advisors. Based on all four reports received, your manuscript could be reconsidered for publication should you be able to incorporate (or adequately respond to) the concerns of Reviewer 1. In particular, that reviewer is requesting that your manuscript only list those formats which are fully supported. In that same comment, the reviewer notes that textures are not supported and that this is a a fatal limitation - please respond to this concern which, if correct, could preclude acceptance That same reviewer notes that for your article to be accurate, you should acknowledge that this is not a ‘complete toolbox’ (but instead limited to a specific list of data visualizations) - again you should respond to this concern

·

Basic reporting

Meets the publication standards with respect to level of English/grammar. Summary of the field and references.

Experimental design

This follows feedback in the first review.

The input format list. The author admits that not all features of all formats are supported. I accept that, it is notoriously problematic. I suggest that if a format is not fully supported then it should not be in the supported list. I would rather see a short list (even if it is just 1) of formats that are known for certain to work irrespective of what features of the format employed. Less is more.

For example I installed MeVisLab 4.8 for Mac, then installed 3D PDF support from here (3rd April)
https://zenodo.org/record/48758
Opened an obj file in the app but it appeared without the associated textures. The revised paper acknowledges that textures are not supported by the tool, this in my opinion is a fatal limitation, noting that it is not a limitation of u3d which does support at least diffuse and transparency textures.

Validity of the findings

This follows feedback in the first review.

The title refers to “a complete toolbox”, I think the author agrees that the proposed solution is not a complete toolbox. The types of data visualisation that is supported by the intersecting feature set of this tool, u3D and 3D PDF is quite narrow.

Additional comments

The author has addressed most of the feedback from the first review in the revised document.

I agree that my view of the software tool discussed in the paper is clouded by my opinion that 3D PDF is not a suitable tool for the interactive representation of data within a paper. I accept that some journals accept it but in my opinion it is not the future. I would freely point out this possible bias to the editors. In my comments here I have tried to only address those points relevant to the proposal in the paper, not to the suitability or otherwise of 3D PDF as a means of representation for data visualisation.

Regarding file size, I agree that 1GB may not be a significant download these days, but I still maintain that the solution is a big hammer to crack a small nut. Note I made a mistake in my earlier review, it is not the download that is 4GB but the final program (actually 5.3GB) on a Mac. That is, it is a lot to download, install and learn for what to me should be a simple task of creating a 3D PDF. In deed the app itself is all that is needed, not clear why it is based upon MeVisLab. I note that 2 of the 3 other reviewers, and perhaps all 3, admit they didn’t try the software. I suggest the hurdle was too high.

The author has provided a slightly more interesting model as an example. There is only one viewer, from Adobe. See supplied image that shows shading errors with the example provided. Is that from the viewer or software proposed in the paper? If one is using such a tool to present scientific data it should be a huge concern that there are these representation errors.

Reviewer 4 ·

Basic reporting

same as for the previous version

Experimental design

same as for the previous version

Validity of the findings

same as for the previous version

Additional comments

The author has modified the manuscript to make the "pure user" life easier - it is appreciated.

---

## Round 0.3 · accepted · Accept

We are happy to accept your revised paper for publication in PeerJ Computer Science.